# Single Cell High Dimensional Analysis of Human Peripheral Blood Mononuclear Cells Reveals Unique Intermediate Monocyte Subsets Associated with Sex Differences in Coronary Artery Disease

**DOI:** 10.3390/ijms25052894

**Published:** 2024-03-01

**Authors:** Nandini Chatterjee, Ravi K. Komaravolu, Christopher P. Durant, Runpei Wu, Chantel McSkimming, Fabrizio Drago, Sunil Kumar, Gabriel Valentin-Guillama, Yury I. Miller, Coleen A. McNamara, Klaus Ley, Angela Taylor, Ahmad Alimadadi, Catherine C. Hedrick

**Affiliations:** 1La Jolla Institute of Immunology, La Jolla, CA 92037, USA; nchatterjee@lji.org (N.C.); kley@augusta.edu (K.L.); 2Department of Medicine, Immunology Center of Georgia, Augusta University, Augusta, GA 30912, USA; rkomaravolu@augusta.edu (R.K.K.);; 3Beirne Carter Immunology Center, University of Virginia, Charlottesville, VA 22904, USAamt6b@uvahealth.org (A.T.); 4Division of Endocrinology, University of California San Diego, La Jolla, CA 92093, USA

**Keywords:** monocytes, antibody-sequencing (Ab-Seq), generalized linear mixed mdel (GLMM), Gensini score (GS), coronary artery disease (CAD)

## Abstract

Monocytes are associated with human cardiovascular disease progression. Monocytes are segregated into three major subsets: classical (cMo), intermediate (iMo), and nonclassical (nMo). Recent studies have identified heterogeneity within each of these main monocyte classes, yet the extent to which these subsets contribute to heart disease progression is not known. Peripheral blood mononuclear cells (PBMC) were obtained from 61 human subjects within the Coronary Assessment of Virginia (CAVA) Cohort. Coronary atherosclerosis severity was quantified using the Gensini Score (GS). We employed high-dimensional single-cell transcriptome and protein methods to define how human monocytes differ in subjects with low to severe coronary artery disease. We analyzed 487 immune-related genes and 49 surface proteins at the single-cell level using Antibody-Seq (Ab-Seq). We identified six subsets of myeloid cells (cMo, iMo, nMo, plasmacytoid DC, classical DC, and DC3) at the single-cell level based on surface proteins, and we associated these subsets with coronary artery disease (CAD) incidence based on Gensini score (GS) in each subject. Only frequencies of iMo were associated with high CAD (GS > 32), *adj.p* = 0.024. Spearman correlation analysis with GS from each subject revealed a positive correlation with iMo frequencies (r = 0.314, *p* = 0.014) and further showed a robust sex-dependent positive correlation in female subjects (r = 0.663, *p* = 0.004). cMo frequencies did not correlate with CAD severity. Key gene pathways differed in iMo among low and high CAD subjects and between males and females. Further single-cell analysis of iMo revealed three iMo subsets in human PBMC, distinguished by the expression of HLA-DR, CXCR3, and CD206. We found that the frequency of immunoregulatory iMo_HLA-DR^+^CXCR3^+^CD206^+^ was associated with CAD severity (*adj.p* = 0.006). The immunoregulatory iMo subset positively correlated with GS in both females (r = 0.660, *p* = 0.004) and males (r = 0.315, *p* = 0.037). Cell interaction analyses identified strong interactions of iMo with CD4^+^ effector/memory T cells and Tregs from the same subjects. This study shows the importance of iMo in CAD progression and suggests that iMo may have important functional roles in modulating CAD risk, particularly among females.

## 1. Introduction

Cardiovascular disease is among the leading causes of morbidity and mortality world-wide [1]. Severe vascular inflammation can be a major factor in cardiovascular pathogenesis and may differ between the sexes [2]. The accumulation of monocytes and macrophages in atherosclerotic plaque is closely linked to plaque progression [3]. Monocytes are classified into three distinct subtypes based on their expression of surface markers CD14 and CD16: classical monocytes (CD14^+^CD16^−^), intermediate monocytes (CD14^+^CD16^+^), and nonclassical monocytes (CD14^−/lo^CD16^+^). Recent studies have suggested that these monocyte subtypes play different roles in human cardiovascular diseases. Both classical monocytes and intermediate monocytes have been associated with increased cardiovascular risk [4,5,6,7,8], whereas increases in nonclassical monocytes have been associated with reduced atherosclerosis, at least in mice [9].

Recent studies of ours and others, using high dimensional immunoprofiling methods, have identified multiple immune cells present in the mouse atherosclerotic aorta, including previously unknown immune cell subclusters [10,11]. In human clinical atherosclerosis, we and others have reported changes in human peripheral blood T and B cells [12,13] and plaque-localized immune cells and macrophages [14,15,16]. We have reported detailed phenotypic heterogeneity in human peripheral blood monocytes from healthy individuals [17]. However, human monocyte and DC heterogeneity in clinical atherosclerosis have not been explored in detail.

The aims of our current study were to define how monocytes and DC differ in human subjects with mild to severe coronary artery disease (CAD) and to assess whether any monocyte or DC subsets were correlated with increased CAD risk. We analyzed phenotypes and transcriptomes of monocytes and DC in peripheral blood cells (PBMC) from 61 subjects using Antibody-sequencing (Ab-Seq) and linked myeloid cell heterogeneity to clinical characteristics of CAD in each subject. The results of our study revealed important differences in monocyte subset frequencies, phenotypes, and gene expression between subjects with low and high CAD.

## 2. Results

Coronary angiography allows us to apply a well-validated quantitative atherosclerosis severity score, known as the Gensini score, that has proven to be highly effective in predicting the risk of future cardiovascular events. Subjects within our study were stratified based on Gensini scores as either having low CAD (Gensini scores ≤ 6) or high CAD (Gensini scores > 32) [12,13]. We included both males and females in our study, and all statistical analyses were adjusted for statin use, smoking, and diabetes as covariates. Relevant clinical details on our CAVA subjects that were used in this study are shown in Figure 1A and illustrate that our cohort was well-matched for all clinical cardiovascular risk factors other than Gensini scores.

Peripheral blood mononuclear cells (PBMCs) from each subject were analyzed for surface proteins and transcriptomes using the BD Rhapsody platform (Figure 1B). In Rhapsody, oligo-tagged antibodies were combined with a custom 487 ‘immune-related’ gene panel, and single-cell sequencing was performed. In this study, we analyzed 133,788 single CD45^+^ immune cells. Using Seurat 4.0 and the Weighted Nearest Neighbor (WNN) algorithm [18], we clustered the cells using both RNA and surface protein expression to generate four main classes of immune cells, including NK cells, T cells, B cells, and monocytes plus dendritic cells (Mo+DC) (Figure 1C). We chose to use the Weighted Nearest Neighbor (WNN) method in the clustering stage, as although myeloid cells are similar to each other in terms of cell surface protein expression, they have differences in terms of RNA expression [18]. The relative information content of the RNA to the protein for each individual cell is termed ‘RNA weights’. RNA weights for major CD45^+^ immune cells are shown (Appendix A) and reveal differences among immune cell types. Heatmaps illustrating scaled and normalized expression of each surface protein marker (Figure 1D and Appendix A) were used to identify CD45^+^ immune cell populations. Detailed analyses of CD4^+^ T cells and B cells from this dataset have recently been reported elsewhere [12,13].

As our primary focus in this study was to determine if myeloid cell frequencies, surface protein markers, or gene expression could be associated with and/or predict CAD severity, we subclustered 47,449 myeloid cells (from the orange cluster in Figure 1B) using the WNN method in an unbiased manner. We obtained six clusters projected on a UMAP that were identified based on protein expression: classical monocytes (cMo), DC3, plasmacytoid DC (pDC), classical DC (cDC), intermediate monocytes (iMo), and nonclassical monocytes (nMo) (Figure 2A). Feature plots of the surface protein expression of CD14, CD16, CD123, and CD11c (Appendix A) and heat maps of protein expression (Figure 2B and Appendix A) were used to identify the clusters. Ridge plots of key identifying markers CD14, CD16, CD123, and CD206 confirmed cMo, iMo, and nMo monocytes, pDC and DC3, respectively (Figure 2C).

We identified the top 100 upregulated (Figure 2D) and downregulated (Appendix A) differentially expressed genes (DEG) in each of the six clusters compared to the other five clusters. DC3, a recently identified inflammatory DC [19], had gene expression profiles more similar to that of classical monocytes rather than to cDC, with the exception of S100A10 and Ly86 gene expression (both of which were upregulated in cDC and DC3 compared to cMo). DC3 showed upregulation of the transcription factor NR4A1, which was likewise elevated in iMo and nMo but not in cMo. We have reported that *NR4A1* is the master transcription factor for the regulation of nonclassical monocyte development [20]. On the other hand, pDC had a unique gene expression profile. Of interest, iMo shared similar gene expression upregulation patterns with nMo rather than cMo (Figure 2D). When examining downregulated genes (Appendix A), we observed similar patterns among cell types as described above for upregulated genes. Figure 2E shows the unique differentially expressed genes in each subset compared to all other subsets. Each of the six clusters has a unique gene profile. The complement genes *C1QA* and *C1QB* were uniquely expressed in iMo. *CCR2* is uniquely expressed in cMo, and *STAT5A* and *SOD2,* among others, are uniquely expressed in nMo.

We utilized Ingenuity Pathway Analysis to infer the functions of these six clusters based on their differential gene expression. For monocytes, cMo (lavender bars) showed activation of pathways leading to cytokine signaling, NFkB signaling, and T cell exhaustion, whereas iMo (black bars) and nMo (green bars) showed activation of quite different signaling pathways, including *NR4A1* signaling, antibody-mediated phagocytosis, phospholipase C signaling, and NK signaling (Figure 2F). Interestingly, cDC (magenta bars) and pDC (orange bars) showed inhibition of these pathways, again supporting their different lineage.

To assess whether frequencies of these six myeloid cell subsets were associated with CAD status in our CAVA subjects, we quantified the proportions of each subset in all 61 subjects and plotted these cell frequencies based on either low CAD (Gensini score ≤ 6) or high CAD (Gensini score > 32). We found that only iMo frequencies were associated with CAD status, with higher iMo frequencies in subjects with high CAD (adjusted *p* = 0.024; Figure 3A). Surprisingly, we found no association of cMo frequencies with CAD status (Figure 3A), even though cMo are reported to have causal effects on plaque initiation [21]. Further, Spearman correlation analysis of myeloid cell frequencies with Gensini scores revealed a significant positive correlation of only iMo with Gensini scores (r = 0.314, *p* = 0.014) (Figure 3B). pDC frequencies tended to positively correlate with higher Gensini scores, but this did not reach significance. Surprisingly, we found trending negative associations of both DC3 and cMo with Gensini scores (Figure 3B).

We next asked whether there were gene expression differences in iMo between subjects with high versus low CAD. Comparison of gene ontology biological processes at the single cell level between iMo from CAVA subjects with low versus high CAD revealed clear and stark differences (Figure 3C). In iMo from subjects with low CAD, pathways related to regulation of innate immunity and inflammatory response, apoptosis, cell migration and adhesion, and signal transduction were elevated; however, in iMo from high CAD subjects, we see the opposite: there was higher activation of pathways leading to regulation of the adaptive immune response.

We identified T cell and B cell subsets in PBMC obtained from the same subjects (Figure 1); analyses of these other immune cell populations in CAVA subjects have been reported [13,22]. Here, we wanted to discern if iMo interacted or communicated with T or B cells. While cell:cell interactions are less prevalent in circulation compared to the vessel wall, we aimed to determine if we could infer any communication between iMo and other immune cells and identify distinctions in iMo communication under low and high CAD conditions. We used the CellChat package to assess the number of interactions and interaction weights of ligand–receptor pairs between iMo and lymphoid immune cells. In the comprehensive analysis involving cells from both high CAD and low CAD groups, our findings revealed that iMo communicated with NK cells, CD4^+^ cells, CD8^+^ cells, and B cells (Figure 4A), although cell:cell interactions were most robust between iMo and CD4^+^ effector/memory cells (defined here as CD4^+^CD45RA^−^CCR7^−/lo^). Key ligand–receptor signaling pairs, in cells from all subjects from both the high CAD and low CAD groups, included iMo PSGL-1 (*SELPLG*) with B cells and naive CD4^+^ and CD4^+^ eff/mem cells, iMo Galectins with naive CD4^+^ and CD4 eff/mem cells, CD8^+^ T cells, and NK cells; and CD86 on iMo with Tregs (Figure 4B). Comparing communication pathways between iMo from low and high CAD subjects revealed more interactions of the GALECTIN, SELPLG, CD48, and CD86 pathways in iMo from high CAD subjects compared to low CAD subjects. These ligand–receptor interactions included Galectin-9 with CD44 and CD45 on CD4+ eff/mem cells, CD48 with CD244A on NK cells, and CD86 with the exhaustion marker CTLA-4 on T regs (Figure 4C and Appendix A). Overall, GALECTIN, SELPLG, CD48, and CD86 signaling interactions were stronger in iMo from high CAD subjects than in subjects with low CAD (Figure 4D).

To gain additional insights into the association of iMo frequencies with CAD, we investigated the iMo subset (black cluster in the UMAP from Figure 2A) at a higher resolution. We identified three subclusters of iMo as displayed on the UMAP (Figure 5A). Importantly, we examined the association between these iMo subsets and CAD status. We found that one of the three iMo subsets, iMo_HLA-DR^+^CXCR3^+^CD206^+^, was significantly elevated (*adj.p* = 0.006) (Figure 5B). Additionally, only the immunoregulatory iMo_HLA-DR^+^CXCR3^+^CD206^+^ subset had a significant positive correlation with Gensini scores (r = 0.367, *p* = 0.004) (Figure 5C).

Heatmaps (Figure 5D and Appendix A) and feature plots (CD14, CD16, HLA-DR, CD86, CD206, CD163, CD45RA; Appendix A) were utilized to identify iMo subsets based on key surface marker protein expression. We identified a novel immunoregulatory iMo subset (iMo_HLA-DR^+^CXCR3^+^CD206^+^), an HLA-DR^+^ subset, and an HLA-DR^int^CCR2^lo^ subset (Figure 5D) in these subjects. We examined the top upregulated DEG among the three iMo subsets (Figure 5E) and found that the iMo_HLA-DR^+^CXCR3^+^CD206^+^ subset uniquely expresses the complement genes *C1QA* and C1QB. The iMo_HLA-DR^int^CCR2^lo^ subset had high *FCER1g* and *LGALS1* expression. The iMo_HLA-DR^+^ subset showed a unique high expression of *NR4A1* and *CX3CR1.* Functional pathway analysis using the AUCell package revealed unique potential functional differences among the three subsets (Figure 5F). The iMo_HLA-DR^+^CXCR3^+^CD206^+^ subset showed upregulation of genes involved in cell activation, migration, and inflammation. The iMo_HLA-DR^int^CCR2^lo^ subset was associated with activation of the adaptive immune response and cell differentiation, while iMo_HLA-DR^+^ appeared to be a possible proliferating subset as it showed high expression of cell cycle pathways, with upregulation of genes including *CCND2, CCNL1, IER3, BCL2, DUSP1,* and *PCNA*.

We next analyzed the pseudotime trajectory of these three iMo subsets compared to cMo and nMo. Setting cMo as the root based on the known conversion of cMo -> iMo -> nMo [23,24], we confirmed that the three iMo subsets were nestled in the middle of cMo and nMo in the diffusion map trajectory (Figure 6A). Computational analysis revealed that the pseudotime order was cMo -> iMo_HLA-DR^+^ -> iMo_HLA-DR^int^CCR2^lo^ -> iMo_HLA-DR^+^CXCR3^+^CD206^+^ -> nMo (Figure 6B). Figure 6C shows key protein expression for all monocytes across the pseudotime trajectory. Expression of the top 75 temporally expressed genes in the monocytes across pseudotime is shown in Appendix A. As the iMo_HLA-DR^+^ subset expressed cell cycle genes, this trajectory analysis supports the notion that this subset is differentiating into iMo_HLA-DR^int^CCR2^lo,^ which is also reflected in the 3D modeling of the trajectory (Appendix A).

Finally, we utilized high dimensional spectral flow cytometry to validate the discovery of the three iMo subsets in a cohort of subjects from CAVA. We were successfully able to identify all three subsets of iMo in 26 CAVA subjects (n = 10 low CAD with GS mean of 2.4 ± 1.9 S.D. and n = 16 high CAD with GS mean of 59.5 ± 28.4 S.D. (Figure 7). Flow cytometry gating strategies (Appendix A) for these human iMo subsets were designed based on surface marker expression from our previous study [25]. Using this simplified flow cytometry gating strategy in our validation cohort, we confirmed that iMo_HLA-DR^+^CXCR3^+^CD206^+^ cells were elevated in high CAD (Figure 7).

We also examined possible sex differences in links of myeloid cell frequencies with Gensini scores. Our cohort contains 44 males and 17 females. Frequencies of iMo were significantly higher (approximately 2-fold) in females with high CAD compared to females with low CAD (Figure 8A), with no differences observed in males. Our findings remained stable and robust when subject to bootstrapping and subsampling techniques despite the variable sample size of females in high CAD and low CAD. These analyses consistently supported the observed sex differences in cluster proportions, reinforcing the validity of our results. Further analysis revealed that iMo frequencies were positively associated with Gensini score in females (r = 0.663, *p* = 0.004), as shown in Figure 8B and Appendix A, whereas there was a trend in males, but it did not reach statistical significance (Figure 8A).

To identify genes present in iMo that are associated with CAD status, we performed machine modeling. The random forest model was trained with the gene expression from randomly selected iMo cells over 30 iterations, and variable importance scores of the genes were calculated and scaled to the range of 0–100. The iMo genes with the highest importance scores predicted by the model for CAD status were *LYZ*, *CD52*, *LGALS1*, *DUSP1*, *NR4A1*, *LGALS3*, *IFITM3*, *S100A10*, *KLF2*, and *S100A9* (Appendix A).

Looking closer to determine if there are sex differences in gene expression within iMo, we examined the importance ranks of genes in iMo between males and females. Indeed, we observed marked differences in gene expression in iMo between males and females (Figure 8C). Notable were differences in the importance rank of key signaling molecules, including transcription factors, alarmins, chemokines, and kinase pathways. For example, the importance ranks of *S100A10*, *CCL5*, the *LYN* kinase, and *FOSB* were higher in males, while the importance ranks of *NR4A1*, *DUSP2*, and *CTSA* (cathepsin A) were higher in females. Taken together, these data suggest that not only are frequencies of iMo different between males and females with CAD but there also appear to be functional gene differences in iMo between males and females.

When we looked closer at possible sex differences in iMo subset frequencies, we found that frequencies of iMo_HLA-DR^+^CXCR3^+^CD206^+^ monocytes were positively correlated with CAD status in both females and males (Figure 8D). iMo_HLA-DR^int^CCR2^lo^ was positively associated with Gensini scores in females but not significantly correlated in males. The third iMo subset, HLA-DR^+^, was not correlated with Gensini in either sex.

## 3. Discussion

Here, we identified and measured monocyte and dendritic cell subsets in the blood of human subjects with low versus severe CAD. Using an Ab-seq single-cell platform, we used oligo-tagged surface antibodies to identify six major types of myeloid cells in the blood of 61 subjects from the CAVA (Coronary Assessment at Virginia) cohort. These included classical, intermediate, and nonclassical monocytes, plasmacytoid DC, classical DC, and a recently identified CD14^+^ DC that we termed DC3 based on published data. We also obtained the expression of 487 immune cell-related genes in each of the myeloid cells and discovered several myeloid gene associations with CAD. Only the frequency of intermediate monocytes was significantly associated with increased CAD in our cohort. Unbiased subclustering of the intermediate monocytes identified three subsets, each with unique functions. Frequencies of two of these subsets were positively correlated with increased incidence of CAD. Importantly, we observed sex differences in iMo, with females with high CAD having higher frequencies of iMo and different iMo gene expression profiles compared to males.

Clinical investigations have reported that elevated levels of both classical and intermediate monocytes are directly correlated with cardiac dysfunction [26,27,28]. Prior studies using conventional flow cytometry to measure monocyte levels in blood reported an increase in iMo frequencies in subjects with cardiovascular disease and showed a strong positive correlation between iMo and Gensini scores [5,6,29]. In patients with unstable angina pectoris, upregulation of intermediate monocytes was associated with plaque rupture [30]. In this study, we found only iMo to be associated with coronary artery disease. Reasons for the variable findings of monocyte associations in CAD among clinical studies include the type of cardiovascular disease (ex: atherosclerosis, ischemia) studied, how disease burden is measured, the size of the cohort studied, cohort attributes (age, sex, race), presence of additional risk factors, including diabetes and obesity, and the method by which the monocytes were identified and quantified. Many of these published clinical studies relied on flow cytometry using only CD14 and CD16 as markers for quantification. The gating of CD14 and CD16 varies among different studies, and other markers are not considered. A study by our group compared both flow cytometry and CyTOF mass cytometry and found that many flow cytometry studies have incorrectly called populations of iMo with only 87% purity, as intermediate monocytes often do not form a discrete subset by flow cytometry gating using solely the monocyte markers CD14 and CD16 [31]. Thus, the use of flow cytometry by different laboratories can lead to variable findings on monocyte associations with heart disease risk. Here, we utilized unbiased high-dimensional approaches with multiple monocyte markers, which avoids the gating differences found in flow cytometry, to definitively determine whether classical or intermediate monocyte populations were associated with CAD.

Nonclassical monocytes have been associated with reduced atherosclerosis in mice [17]. However, we observed no significant association between nonclassical monocyte frequencies and CAD in our current human cohort. Nonclassical monocytes in humans only account for ~10–20% of the monocyte pool, whereas in mice, the nonclassical monocyte subset accounts for approximately ~40% of total monocytes, suggesting that the monocyte compartment may be regulated differently in mice. Further studies using larger human cohorts may aid in discerning links between nonclassical monocytes and coronary heart disease.

A key feature of our analytical approach is the use of a Generalized Linear Mixed Model (GLMM). Unlike traditional statistical methods, GLMM is specifically designed to account for fixed and random effects. This makes it particularly appropriate for situations involving multiple variables that can affect relationship outcomes, commonly referred to as random effects or confounders. In our dataset, clinical conditions such as CAD status, sex, statin treatment, and diabetes status were considered random effects. When examining the myeloid proportion in relation to CAD status, the random effects included sex, statin treatment, and diabetes status. Similarly, when assessing the association between sex and CAD status with myeloid proportions, the random effects were statin treatment and diabetes status. We also considered patient_id to be a random effect in all the comparisons.

Using machine learning, several genes ranked by their highest importance in iMo were linked to increased CAD. These included the genes *LYZ*, *CD52*, *LGALS1*, *DUSP1*, *IFITM3*, *NR4A1*, *S100A9/10*, and *KLF2*. For example, *LYZ* (Lysozyme) has been linked to vascular inflammation and the progression of atherosclerosis [32]. A variation in the *CD52* gene has been associated with a higher risk of myocardial infarction [33]. Immune cell adhesion is regulated by both Galectin-1 (*LGALS1*) and *LGALS3* (Galectin–3) [34]. Notably, inflammatory conditions are strongly associated with elevated Galectin-1 expression. Galectin–3 is well associated with inflammation and vascular disease [35]. Serum levels of Galectin–3 have been positively associated with Gensini scores [36], and studies have linked Galectin-3 in monocytes to impaired host defense [37]. The enzyme *DUSP1* (Dual-specificity phosphatase 1) has been implicated in the maintenance of blood vessel function [38]. We have previously reported that the transcription factor *NR4A1* regulates nonclassical monocyte development and is necessary for inhibiting *NFkB* activation in monocytes and macrophages in atherosclerosis [20]. The interferon-induced transmembrane protein 3 (*IFITM3*) gene codes for interferon genes that are responsible for regulating immune responses. The calcium-binding proteins *S100A9* and *S100A10* play an important role in inflammation and cell proliferation. Increased expression of *S100A10* is associated with thrombosis and atherosclerosis [39]. The flow-regulatory *KLF2* (Krüppel-like factor 2) transcription factor helps to maintain blood vessel homeostasis, and we have shown that *KLF2* regulates monocyte development [40,41]. Thus, there are gene expression changes in iMo from subjects with low versus high CAD that likely dictate distinct functions. From the gene ontology biologic pathway analysis in Figure 3C, we show the striking observation that the regulation of the adaptive immune response is a clear component of iMo function in only those subjects with more severe, advanced CAD. In contrast, the innate immune response is highly active in iMo from subjects with low or mild CAD. This is the first clear observation of differences in intermediate monocyte function and heterogeneity in humans with mild versus severe CAD.

Interestingly, we observed sex differences in frequencies of iMo in subjects with CAD, with iMos significantly elevated in females with high CAD. All women in our study were postmenopausal, and none were on hormone replacement therapy. Indeed, there is a significant positive association with iMo frequencies and Gensini scores in women but not men (Figure 8B and Appendix A). Using machine learning modeling, we identified several genes in iMo that exhibited sex-based differences in their ability to distinguish between high and low CAD status (Figure 8C). Genes with higher importance (lower rank) in females included *Vmo1*, the phosphatase *Dusp2*, the viral inhibitor *apoBec3g,* and the transcription factor *Nr4a1*. Genes with higher importance in males included the chemokine *CCL5*, the activation marker *CD44*, the kinase *LYN*, and the transcription factor *FOSB*. These data suggest that iMos functionally may be different in males and females, and changes in their signaling pathways may contribute to CAD incidence.

We observed a monocyte/DC subset, which we termed DC3 based on the surface markers reported by Dutertre et al. [19]. This subset clustered close to both classical monocytes and DC (Figure 2A) and had a gene expression pattern like both classical monocytes and cDC (Figure 2D). This DC3 subset expressed *S100A10* and *LY86*, like cDC, yet showed differential expression of many cMo genes. Surface proteins that mark this subset include CD206, CD14, CD33, CD36, and CD11c (Figure 2B and Appendix A). Dutertre and colleagues reported that DC3 arises in response to inflammation and is derived from a DC lineage [19]. Trajectory analyses in our study suggested that this DC3 is developmentally derived from monocytes; however, we did not perform lineage-tracing studies, so further investigation is needed to confirm this hypothesis. In terms of relation to CAD incidence, this DC3 population showed no differences in frequency between individuals with low and high CAD nor were there sex differences in DC3 frequencies in our cohort.

Using computational methods, we identified three subsets of intermediate monocytes in our CAVA cohort. Pseudotime trajectory analysis suggests that HLA-DR^+^ monocytes are the earliest iMo to arise in the periphery (Figure 6A,B). Both gene expression data (Figure 5E) and trajectory analyses (Figure 6B) suggest that these cells are developing into HLA-DR^int^CCR2^lo^ monocytes (Appendix A). This is the first report on trajectory analysis of human iMo subsets. We did not observe striking differences in pseudotime trajectories of iMo between low and high CAD subjects.

Two iMo subsets with similar markers have been previously identified by our group in studies of human non-small cell lung cancer (NSCLC) cancer using CyTOF [25]. A recent study by our collaborators reported the presence of three intermediate subsets expressing similar markers in women with HIV [22]. One of these iMo subsets identified by Vallejo et al. also expressed the chemokine receptor CXCR3 [22], which we termed as ‘immunoregulatory monocytes’ in an earlier study [25]. Whether CXCR3 either serves as a simple biomarker for the association of iMo with CAD or has functional roles in iMo for CAD development is not yet clear. However, levels of the ligands for CXCR3 (CXCL9, CXCL10, and CXCL11) have been localized to human atherosclerotic plaques, and CXCL10 has been associated with the severity of CAD in humans [42]. Moreover, inhibition of CXCL10 using a neutralizing antibody resulted in more stable plaques in mice [43,44,45]. Future studies will assess the role of CXCR3 in iMo in CAD.

Serum C1Q levels have been associated with increased cardiovascular risk in humans [46,47]. C1Qa and C1Qb were both uniquely expressed in iMo in our cohort (Figure 2E) and were uniquely linked to the immunoregulatory iMo_HLA-DR^+^CXCR3^+^CD206^+^ subset (Figure 5E), suggesting that this subset functionally may influence the complement cascade. C1Q functions as a direct link between innate and adaptive immunity [48]. Again, the link of complement with this immunoregulatory CXCR3^+^ iMo subset may be important functionally in CAD progression.

We validated our findings in those CAVA subjects for whom additional PBMC was available using spectral flow cytometry. We did confirm the elevations in high CAD subjects of the iMo_HLA-DR^+^CXCR3^+^CD206^+^ subset (Figure 7). However, due to the lower numbers of subjects, coupled with the known variability in blood leukocyte values among human subjects, we were unable to confirm a difference between CAD status and iMo_HLA-DR^int^CCR2^lo^ frequencies. Unfortunately, we did not have enough males and females in each group to perform a statistical comparison between the sexes. Even so, the generation of such a flow cytometry gating strategy (see Appendix A) will be highly useful for the simple quantification of these intermediate monocytes in the future for clinical cardiovascular studies.

A limitation of our current study is the sequencing panel that we used consists of only 487 genes. These are well-known immune response genes, so we obtained important information about immune response differences in iMo between groups in our cohort (see Figure 3C and Figure 5F), but we are missing transcriptomic information from other non-immune-focused pathways. Future studies will expand this cohort using single-cell whole transcriptome approaches.

In summary, we conclusively show that intermediate monocytes are elevated in human subjects with clinically high CAD and are strongly correlated with Gensini scores. Not only are there differences in the frequency of iMo, but there are also significant immune-related gene differences between iMo in subjects with clinically low versus high CAD. We found striking differences in gene expression in iMo from females versus males. Moreover, we identified three intermediate monocyte subsets that have distinct functions, including an immunoregulatory iMo subset that is associated with high CAD that expresses the CXCR3 receptor as well as C1Q complement genes. Our study indicates the importance of iMo in CAD progression and suggests that iMo may not only have a possible predictive utility for CAD risk, particularly in females but also have important functional roles in mediating CAD progression. Furthermore, our observation of iMo frequency elevations in females with high CAD compared to females with low CAD could become an important aspect for directing personalized cardiovascular medical care in the future.

## 4. Methods

### 4.1. Human Subjects

We evaluated 65 subjects, ranging in age from 42 to 78 years old and matched for cardiac risk factors, from the Coronary Assessment in Virginia (CAVA) cohort. These participants, referred by their physicians for invasive coronary artery evaluation, were recruited through the Cardiac Catheterization Laboratory at the University of Virginia Health System in Charlottesville, VA, USA [13,22]. Prior to enrollment, all participants provided written informed consent, and the study received approval from the University of Virginia Human Institutional Review Board (IRB No. 15328). Peripheral blood mononuclear cells were obtained from the participants before undergoing coronary angiography. The procedures were followed according to the regulations established by the Clinical Research and Ethics Committee and the Helsinki Declaration of the World Medical Association. Clinical features, including Gensini score, age, sex, hsCRP, serum lipids, statin treatment, HgbA1c, and diabetes status, are listed in Figure 1A.

### 4.2. Quantitative Coronary Angiography (QCA)

The methods for performing Quantitative Coronary Angiography (QCA) and the calculation of the Gensini score have been outlined in detail in studies conducted by Saigusa et al. [12]. and Pattarabanjird et al. [13]. We obtained PBMCs from 61 subjects who are enrolled in the University of Virginia Coronary Assessment at Virginia (CAVA) cohort. Based on QCA, subjects with a Gensini score > 32 were classified as having high (severe) CAD, and subjects with a Gensini score ≤ 6 were classified as having little to no CAD. These subjects were all undergoing medically necessary coronary angiography.

### 4.3. PBMC Sample Preparation for Antibody-Seq

Peripheral blood was collected from individuals with CAD and individuals who underwent coronary angiography to exclude CAD. A total of 65 PBMC samples were examined, and 61 samples passed quality control with cell viability > 80%. The blood was drawn into BD K2 EDTA vacutainer tubes and processed at room temperature within one hour of collection. PBMCs were isolated by Ficoll-Paque PLUS (GE Healthcare Biosciences AB, Uppsala, Sweden) gradient centrifugation using SepMate-50 tubes (Stemcell Technologies Inc., Vancouver, BC, Canada) according to the manufacturer’s protocol. Trypan blue staining of PBMCs was performed to quantify live cell counts. The isolated PBMCs were cryopreserved in a freezing solution (90% FBS with 10% DMSO). PBMC vials were stored in Mr. Frosty (Thermo Fisher, Waltham, MA, USA) for 48 hrs at −80 °C and subsequently transferred to liquid nitrogen for long-term storage. To minimize batch effects, eight samples were processed together on the same day, thawed in a 37 °C water bath, and centrifuged at 400× *g* for 5 min, and pellets were resuspended in a cold staining buffer. The viability and cell count of each tube were assessed using the BD Rhapsody Scanner. The tubes were then centrifuged at 400× *g* for 5 min and resuspended in a cocktail of 49 AbSeq antibodies (2 μL each and 20 μL of SB) on ice for 30–60 min following the manufacturer’s recommendations, then washed and counted again. Each subject’s cells were tagged using a Sample Multiplexing Kit (BD Biosciences, San Jose, CA, USA) containing oligonucleotide cell labeling. The cells were washed thrice, mixed, counted, stained with the 49-antibody mix, washed thrice again, and loaded onto Rhapsody nanowell plates (four samples per plate).

### 4.4. Library Preparation

The primed plate was filled with cells at a concentration of 800–1000 cells/μL. To initiate reverse transcription, the plate was placed on a thermomixer and incubated at 37 °C and 1200 rpm for 20 min. Afterward, Exonuclease I was added to the plate and further incubated on the thermomixer for 30 min. The plate was then transferred to a heat block and incubated at 80 °C for 20 min. The cDNA library was prepared using the BD protocol, following the instructions provided by Vallejo et al. Finally, the quality control and quantification assessments of the cDNA library were conducted using the TapeStation, Qubit kits, and associated reagents (Thermo Fisher, Waltham, MA, USA).

### 4.5. Sampling

The BD recommended sequencing depths for the pooled samples were as follows: 40,000 reads per cell for Ab-Oligos sequencing, 20,000 reads per cell for mRNA sequencing, and 600 reads per cell for Sample Tags. Consequently, a total of 60,600 reads per cell were obtained for sequencing on the NovaSeq platform. The pooling and sequencing depth specifications, as well as the number of cells loaded on each plate, were optimized for S1 and S2 100 cycle kits provided by Illumina. After the sequencing process was completed, the resulting FASTA file and FASTQ files from the NovaSeq were uploaded to the Seven Bridged Genomics pipeline. In this pipeline, the data were filtered and organized into matrices and CSV files. The DoubletFinder package in R [49] was utilized to eliminate doublets, and cells with less than 128 sequenced antibody molecules were excluded. Antibody sequencing data underwent CLR (centered log-ratio) normalization. Two of the 51 antibodies had very low detection and were excluded from all further analyses. Additionally, all transcripts were normalized based on the total UMIs (Unique Molecular Identifiers) in each cell and multiplied by a scale factor of 10,000.

### 4.6. Seurat Workflow for Targeted Data

The analyses were conducted using R (version 4.1.0) and the Seurat v4 package. The “vst” method was employed to identify the top 200 RNA variable features. All 49 ADT features were included in the analysis. The resulting matrices were then scaled, and Principal Component Analysis (PCA) was performed. From the 50 principal components generated, the first 20 were chosen for subsequent batch correlation analysis. For this analysis, the Harmony package v0.1.1 [50] was utilized. The batch-corrected data were used in the ‘FindMultiModalNeighbors’ function to identify nearest neighbors based on the Weighted Nearest Neighbor (WNN) technique. The results were used for subsequent analyses, including clustering and visualization. The Louvain clustering method [51] was employed with a resolution of 1 to identify cell clusters. A WNN-UMAP (Uniform Manifold Approximation and Projection) was created using the ‘RunUMAP’ function, with parameters set as spread = 0.3, n. neighbors = 50, and set.seed = 42. Clusters containing fewer than 100 cells were excluded from further analysis. For the clustering of myeloid cells, normalization was performed on both modalities, followed by PCA. Markers that were not stained in myeloid cells were excluded, and ‘Harmony’, ‘FindMultiModalNeighbors’, and ‘FindClusters’ were rerun with a resolution of 1.7, resulting in a total of 46 clusters. After filtering and merging clusters, the clusters were grouped and annotated into myeloid immune cell types using marker genes and visualization techniques such as UMAPs, heatmaps (at both cell-level and average expression), and feature plots were used.

### 4.7. Differentially Expressed Gene (DEG) Analysis

DEG analyses were performed using the Seurat v4 package. To identify these DEGs, all genes were examined. However, certain genes, namely HLA.A, HLA.B, HLA.DMA, HLA.DPA1, HLA.DQB1, HLA.DRA, TRAC, and GAPDH, were excluded due to their high variability among patients. Within each cluster, the ‘FindAllMarkers’ function was employed with a logfc.threshold of 0.58 and a min.pct of 0.25. Genes that exhibited an adjusted *p*-value < 0.05 were deemed significant. Downregulated genes were characterized by average log2FC values < 0, while upregulated genes had average log2FC values > 0. Unique DEGs were observed exclusively in one cluster.

### 4.8. Ingenuity and AUCell Pathway Analyses

The biological functions of differentially expressed genes in the major clusters were investigated using QIAGEN’s Ingenuity Pathway Analysis [52] (IPA, version 01-21-03). For IPA analysis, a ‘logfc.threshold’ of 0.25 was employed to identify the differential expressed genes. Canonical pathway analysis was performed on gene lists. From each dataset, IPA was used to predict individual signaling pathways. To determine whether our dataset and the IPA references differed significantly, Fisher’s exact tests were used. A reference set of ‘genes only’ and a species of interest of ‘human species’ were included in IPA settings. Pathways considered were only those with a z-score ≥ 2 (predicted activation), ≤ −2 (predicted inhibition), and a *p*-value < 0.05 (derived from Fisher’s exact test).

Given the higher similarity among the monocyte subsets, the AUCell R package version 1.16.0 was employed to investigate the Gene Ontology (GO) Biological Processes at the cellular level. This approach enables uncovering finer distinctions between these subsets. Only the gene sets with at least 10 genes present in our dataset were included in the analysis. The gene sets were downloaded from the GSEA database.

### 4.9. CellChat Analysis

The R package CellChat v1.6.1 [53] was used to predict and compare interactions between the immune cell types and the CAD status. This package utilizes a combination of known molecular interactions and gene expression levels to estimate the likelihood of biological interactions occurring. CellChat used the normalized gene expression matrix to generate the CellChat object, configure the ligand–receptor interaction database for validation, and preprocess the expression data to facilitate cell communication analysis. During the analysis, the ‘filterCommunication’ function was applied with a min.cell of 10. The iMo cluster was specified as the source, while all other immune cell types were designated as targets.

### 4.10. Diffusion Pseudotime

Trajectory analyses were conducted using the Destiny v3.8.1 package in R [54]. To facilitate visual comparison, the Seurat object was downsampled based on the CAD status. An expression matrix was created, comprising normalized counts of the top 200 RNA variable features and 49 ADT features for all myeloid annotations. This matrix served as input for the ‘DiffusionMap’ function, generating diffusion maps with a local scale parameter sigma set to ‘local.’ Diffusion pseudotime was calculated using the ‘DPT’ function with default settings and a chosen tip cell. Branches were identified using the ‘branch_divide’ function. Temporally expressed genes or markers were determined by regressing each gene or marker against the pseudotime variable by applying the Generalized Additive Model (GAM) using the gam R package v1.22 [55]. The top 75 genes and top 49 markers with the most significant time-dependent model fit were visualized using the ‘Heatmap’ function from the ComplexHeatmap package v2.10.0 [56]

### 4.11. Random Forest Machine Learning Algorithm

The R packages, including Caret version 6.0-92 [57] and randomForest version 4.7-1.1 [58], were used to calculate variable importance scores for distinguishing between High CAD and Low CAD conditions through machine learning models. The dataset consisted of single-cell marker expression data and two categories: High_CAD and Low_CAD. ML models were trained using the caret package ‘train’ function, performing 30 iterations. In each iteration, 996 cells were randomly chosen from each group according to the number of cells in the smaller group, resulting in an equal number of cells in each group. Prior to the training process, the data were scaled and centered. The resampling method was set to ‘repeatedcv’, with 10 folds, 10 repeats, and tune length of 10. The variable importance was estimated and scaled using the ‘varImp’ function from the caret package.

### 4.12. Flow Cytometry

PBMCs were isolated from blood samples from 31 subjects from CAVA using Ficoll-Paque PLUS (GE Healthcare Biosciences AB, Uppsala, Sweden) gradient centrifugation. PBMCs were cryopreserved in a freezing solution (90% FBS with 10% DMSO) in liquid nitrogen until use. Frozen PBMCs were thawed, and 10 mL 1XPBS was added to each sample, followed by centrifugation at 400 g for 10 min at RT. Viability and cell count were assessed by both a hemocytometer and a cellometer (Nexcelcom, Lawrence, MA, USA). 1.5 × 10^6^ cells were added into 96 well plates, incubated with a Human TruStain FcX™ (Fc Receptor Blocking Solution; Biolegend, San Diego, CA, USA) for 10 min at 4 °C followed by fixable viability dye to exclude dead cells (Live Dead Blue, Invitrogen, Waltham, MA, USA) for 30 min at 4 °C, washed and antibody master mix against surface markers were added for 40 min at 4 °C. Anti-CD3, Anti-CD19, and Anti-CD56 were used to exclude T-cell, B-cell, and NK cell populations, respectively. CD14^+^CD16^−^, CD14^+^CD16^+^, and CD14^−/lo^CD16^+^ were defined as classical monocytes (cMo), intermediate monocytes (iMo), and nonclassical monocytes (nMo), respectively. Antibodies used in this study are listed in Appendix A. Data were acquired using Cytek Aurora 5 lasers Spectral Flow Cytometer (Cytek Biosciences, CA, USA) and analyzed with FlowJo software (BD Biosciences, San Jose, CA, USA). Fluorescence-minus-one (FMO) controls from healthy donors were used to set the gates. 

### 4.13. Statistical Analysis

For the clinical table and flow cytometry validation data, Prism software (GraphPad, version 10.1.0) was used for statistical analysis. A nonparametric two-tailed Mann–Whitney U test was used to determine the differences between two groups of patients (high CAD vs. low CAD). A significance level of *p* < 0.05 was considered statistically significant. Generalized Linear Mixed Model (GLMM) tests were utilized to examine the significance of changes in myeloid proportions for BD Rhapsody sequencing using the lme4 package v1.1-31 [59]. Various conditions, including CAD status, sex, statin treatment, and diabetes status, were considered. The GLMM model was modified based on the specific conditions under investigation. Up to four clinical variables were considered as potential confounders and treated as random effects. Statistical significance was determined based on an adjusted *p*-value < 0.05. Details regarding the specific tests employed can be found in the figure legends. To address the variable sample size of females between high CAD and low CAD compared to males in our dataset, we employed bootstrapping and subsampling techniques. Bootstrapping involved resampling with replacement, while subsampling matched the female cohort’s size in different CAD statuses. These methods aimed to assess the robustness of our findings regarding sex-related differences in cluster proportions. Additionally, Spearman correlations were performed to assess the relationship between myeloid proportions and Gensini score using the Hmisc package v4.7-2 [60]. A significance level of *p* < 0.05 was considered for the correlations. Validation data were analyzed by Mann–Whitney U-test using GraphPad prism (version 10.1.0).

## Figures and Tables

**Figure 1 ijms-25-02894-f001:**
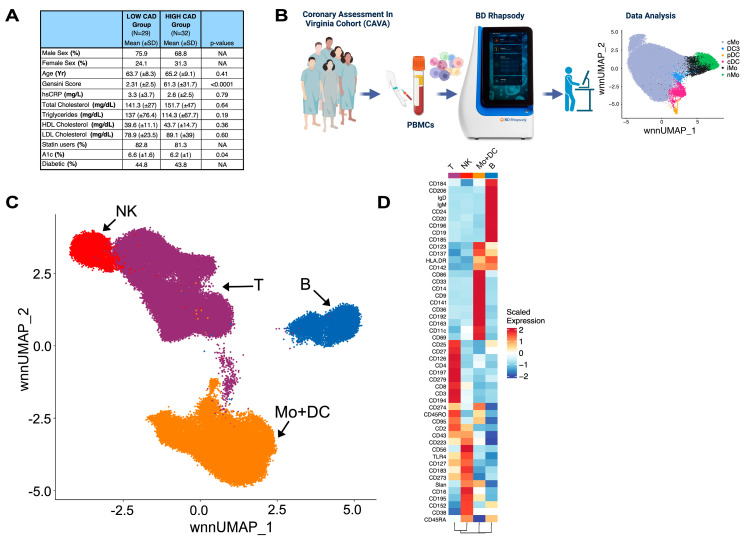
Study Design and CD45^+^ Immune Cell Identification in PBMC of CAVA subjects. (**A**) Clinical parameters of 61 subjects in CAVA. Significant differences were tested using the Mann–Whitney U test. Gensini Score (*p* = < 0.0001) was statistically significant between the CAD groups. (**B**) Experimental scheme of study. Figure generated with BioRender (https://biorender.com/, accesed on 1 January 2024) (**C**) Weighted nearest neighbors (WNN)-based UMAP of 133,788 single CD45^+^ immune cells in PBMC of all study subjects in CAVA. (**D**) Scaled expression heatmap of surface proteins of major immune cell types identified in Panel C.

**Figure 2 ijms-25-02894-f002:**
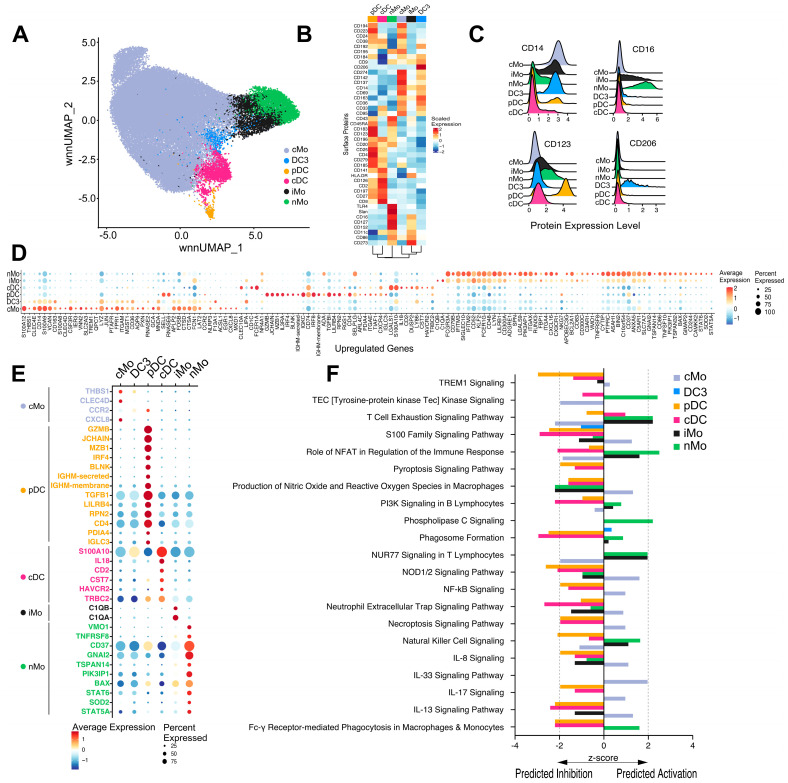
Six major myeloid cell phenotypes present in CAVA subjects. (**A**) WNN-based UMAP of myeloid subclustering of 47,449 single myeloid cells from the Mo+DC cluster in Figure 1C. (**B**) Scaled expression heatmap of surface proteins used to identify myeloid subsets in Panel A. (**C**) Ridge plots comparing surface protein expression of primary markers CD14, CD16, CD123 and CD206 in each of the six myeloid clusters. (**D**) Bubble plot of average expression of the top 100 upregulated differentially expressed genes in each of the 6 myeloid subsets from Panel A. Size of bubble represents percent of cells expressing each gene shown. (**E**) Bubble plot of average expression of unique genes identified in each of the 6 subsets, comparing each subset to the other. Gene names are colored by cell type in Panel A. Size of bubble represents percent of cells expressing each gene shown. (**F**) Genes differentially expressed among six major myeloid cell clusters (cMO, iMo, nMo, cDC, pDC, and DC3) were analyzed using Ingenuity Pathway Analysis. Differentially expressed genes calculated using log2FC were submitted for pathway analysis. Activation z-scores of key immune cell functions are shown in the bar graphs cMo (lavender bars), DC3 (blue bars), pDC (orange bars), cDC (magenta bars), iMo (black bars), and nMo (green bars). The bar visualizes the activation z-score for pathways that have been predicted to be activated (a positive z-score) or inhibited (a negative z-score).

**Figure 3 ijms-25-02894-f003:**
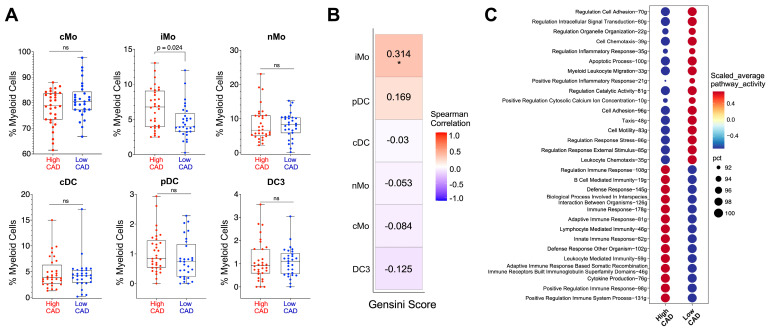
Intermediate CD14^+^CD16^+^ monocytes are higher in CAVA subjects with CAD. (**A**) Frequencies of the six subsets of myeloid cells that were identified in Figure 2. Frequencies are shown as percentage of total myeloid cells. High CAD subjects are shown in red; low CAD subjects are shown in blue. Each dot is an individual subject. iMo were significantly different using the Generalized Linear Mixed Model (GLMM), where sample, sex, diabetes, and statin were considered as random effects. (**B**) Spearman correlation of frequencies of each myeloid subset with Gensini scores of each subject. Red suggests positive correlation with CAD; blue suggests negative correlation with CAD. *: represents significance at 0.05 level (**C**) Single-cell level comparison of GO Biological Processes between High CAD and Low CAD using the AUCell method. The bubble plot illustrates the scaled average pathway activity, while ‘pct’ represents the percentage of cells within each cluster and pathway that had recorded values. ns: non-significant.

**Figure 4 ijms-25-02894-f004:**
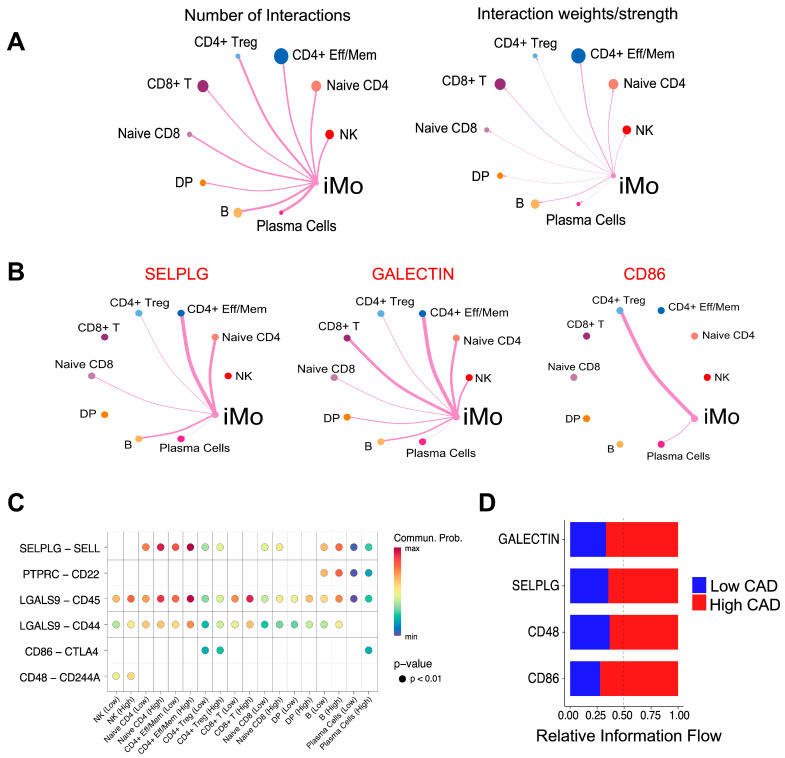
Cell communication of iMo with other CD45^+^ immune cells in PBMC from subjects in CAVA. (**A**) The CellChat package was used to identify the number of interactions and interaction weights/strengths of all iMo with other immune cells in the blood. The iMo were designated as ‘senders’ and other immune cells as ‘receivers’ in these analyses. (**B**) Key communication pathways identified for iMo communication to other immune cells, including SELPLG (P selectin glycoprotein ligand-1), GALECTIN (Galectins), and CD86. (**C**) Bubble plot illustrating key ligand–receptor pairs from pathways in Panel B that demonstrate increased signaling in High CAD. The color shows communication probability. Dots are shown only if the *p*-value for significant interaction was *p* < 0.01. The x-axis shows each receiving cell type in low CAD (Low) or High CAD (High) for comparison. (**D**) Stacked bar plot illustrating the overall information flow for each signaling pathway, as determined by the sum of communication probability within the inferred network among both Low and High CAD groups. Blue represents Low CAD and red represents High CAD. Y-axis shows the only the significant signaling pathways; relative information flow is defined on a scale of 0 to 1 on the x-axis. The dotted line is at 0.50 for comparison.

**Figure 5 ijms-25-02894-f005:**
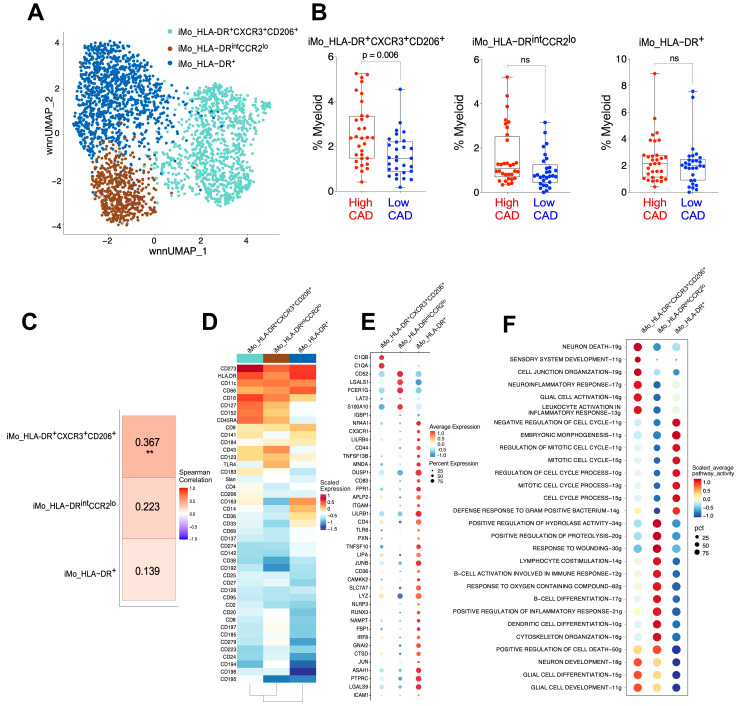
Identification of three novel iMo subsets in PBMC of CAVA subjects. (**A**) The black subset of cells (iMo) in Figure 2A was subclustered, and 3 subsets were identified and shown in the WNN-based UMAP. (**B**) Frequencies of the 3 iMo subsets identified in Figure 5A. Frequencies are shown as the percentage of total myeloid cells. High CAD subjects are shown in red; low CAD subjects are shown in blue. Each dot is an individual subject. iMo subsets were significantly different using the GLMM test where sample, sex, diabetes, and statin were considered as random effects. (**C**) Spearman correlation of frequencies of each iMo subset with Gensini scores of each subject. ** represents significance at 0.01 level (**D**) Scaled expression heatmap of surface proteins used to identify the 3 iMo subsets. (**E**) Bubble plots showing average expression of uniquely expressed genes in each of the 3 subsets. Size of bubble represents percent of cells expressing each gene shown. (**F**) Analysis of GO Biological Processes activity at the single-cell level using the AUCell tool. The figure illustrates the scaled average pathway activity, while ‘pct’ represents the percentage of cells within each cluster and pathway that had recorded values. ns: non-significant.

**Figure 6 ijms-25-02894-f006:**
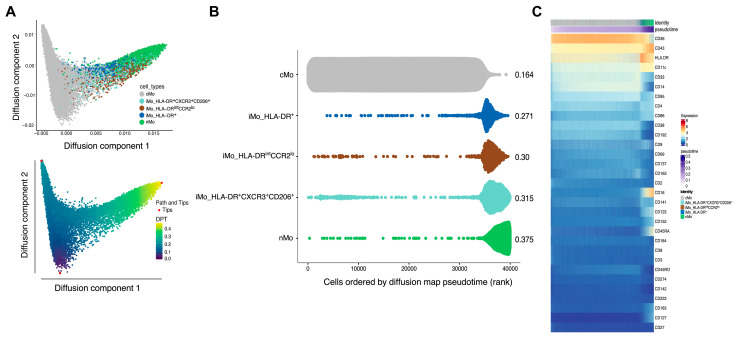
Pseudotime trajectory analysis of 3 intermediate monocyte subsets. (**A**) Trajectory analyses of identified monocytes via diffusion map pseudotime. Each dot represents an immune cell. The transition probability of a diffusion process via pseudotime along the same trajectory map is shown in the panel at the bottom. The three red dots represent possible tips or paths of transition. (**B**) Scatter plot of cells ordered by diffusion map pseudotime. The x-axis shows the rank of each cell ordered by diffusion map pseudotime; The values on the right side of the plot represent the median pseudotime for each cell type. (**C**) Cell-level heatmap of marker expression through the pseudotime. Reading from left to right, the top annotation bar shows the order of cells according to pseudotime; the bottom annotation bar shows the pesudotime gradient.

**Figure 7 ijms-25-02894-f007:**
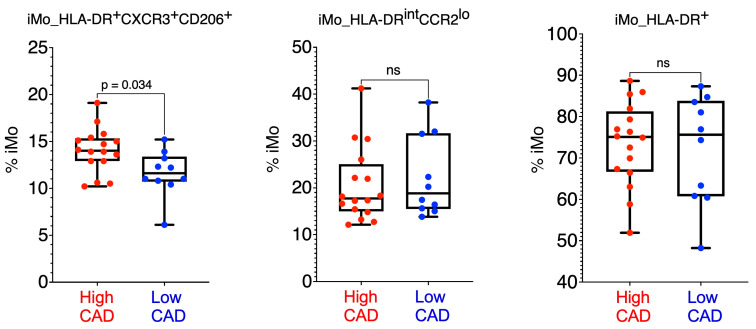
Validation of Ab-seq data by spectral flow cytometry. Using surface protein markers identified in Figure 5, we validated each of the 3 iMo subsets in PBMC of 26 of the original CAVA subjects. Frequencies are shown as percentage of total iMo. Each dot is an individual subject. iMo_HLA-DR^+^CXCR3^+^CD206^+^ is significantly different, *p* = 0.034, Mann–Whitney U-test. ns: non-significant.

**Figure 8 ijms-25-02894-f008:**
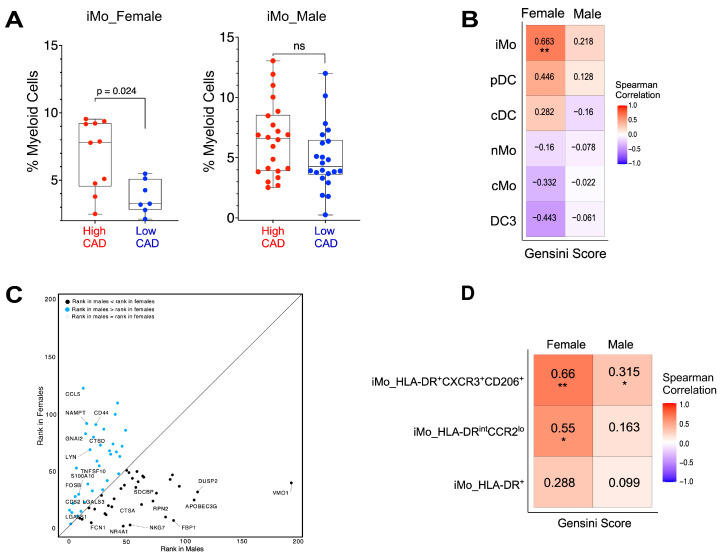
Frequencies of iMo subsets revealed sex-specific differences and are elevated in females with high CAD. (**A**) Frequencies of iMo in males and females with low or high CAD. Frequencies are shown as percentage of total myeloid cells. High CAD subjects are shown in red; low CAD subjects are shown in blue. Each dot is an individual subject. The GLMM was used to test for significance where sample, diabetes, and statin were considered as random effects. (**B**) Spearman correlation of frequencies of each myeloid subset comparing males versus females with Gensini scores. (**C**) Differences in gene rankings between males and females, in their capacity to differentiate between high CAD and low CAD. Labels denote genes with greater than 3-fold variations between males and females. (**D**) Spearman correlation of frequencies of each iMo subset (from Figure 5) with Gensini scores, comparing males versus females. * and ** represent significance at 0.05 and 0.01 levels, respectively. ns: non-significant.

## Data Availability

Data contained within the article.

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
