# Peer review of "Single Cell High Dimensional Analysis of Human Peripheral Blood Mononuclear Cells Reveals Unique Intermediate Monocyte Subsets Associated with Sex Differences in Coronary Artery Disease"

_ijms, 2024, doi:10.3390/ijms25052894_

Round 1
Reviewer 1 Report
Comments and Suggestions for Authors
Chatterjee et al. describe the exploration of monocytes and DC phenotype in 61 subjects who beneficiated angiographic coronary evaluation and were stratified according to their Gensini score. The study is based on mixed single cell transcriptomic (487 immune related genes) associated to surface specific markers (49). They highlight the significant positive association between iMo frequency and “disease” severity. This correlation was due to female population which displayed highly significant correlation contrary to males. Further, the authors identified subsets of iMo according to HLD-DR, CXCR3, CD206, the triple positivity being associated with increased stenosis risk. In iMo from subjects with lower Gensini score (<32), pathways related to regulation of innate immunity and inflammatory response were elevated while in iMo from subjects with higher Gensini score (>32) there was overrepresentation of pathways associated with adaptive immune response. The distribution/frequency of iMo has already been studied in several “atherosclerotic” human cohort. The present article implements these data with transcriptomic analyses, which give insights on cellular and molecular pathways involved. The work and in silico exploration are rather impressive. The paper is quite clear and well written. I would have the following remarks.
1- The terminology “cardiovascular disease” is vague and may be a too large definition for the patient involved in this study who are defined by coronary stenosis measured by Quantitative Coronary Angiography. This should be replaced by coronary heart disease.
2- Since the authors show i) a significant effect of sex on the correlation between iMo and coronary stenosis severity ii) some specific subsets on genes expressed in male vs female iMo (pointing on sex specific pathogenic mechanisms), it is not clear why the authors continue to explore the mixed population (males and females) from the results they obtained in the figure 3. Could it be possible that the absence of significance for the males would be due to insufficient N? in any case males and females should be analyzed separately.
3- In this study, only the frequency of iMo was significantly associated with aortic stenosis severity. This is different from previously obtained results. The role on specific monocytes subsets in inflammatory process and in particular in the pathogenesis of atherosclerosis associated to acute coronary syndrome has been reviewed (25997925). Some data point on a relationship with the amount of circulating classical monocytes and the severity of the disease (29017104), in particular in patients presenting ischemic heart disease (33854919). Others evidence correlation between intermediate monocytes (CD14++, CD16+) and atherosclerotic disease manifestations (22999728, 22785372, 37161473, 27991581). The nature of the cohort and the criteria to stratify disease severity might originate differences observed and reported by several groups. This point should be discussed.
Minor points:
The abstract is a systematic description of experiments and correlative relationships.
It lacks an opening on the perspectives such an important work opened up.
Once the abbreviation introduced “CVD”, it should be used instead of cardiovascular disease L 61, 344, 360, 493, 494, 514.
Reviewer 2 Report
Comments and Suggestions for Authors
Dear Authors,
I was delighted to review your interesting and well-written article. This research looks into monocyte and dendritic cells subsets in coronary heart disease patients. They try to find a link between specific subsets and the severity of atherosclerosis and cardiovascular disease risk. They included 65 subjects and analyzed these cells’ phenotipes and gene expression using a 487 gene panel and single cell sequencing. The severity of the atherosclerosis process was verified by imaging methods- angiography, which validates scientific soundness. The research topic is important to the field of atherosclerosis research and brings one significant new piece of information, yet uncovered, as far as my knowledge goes, by the existing literature: intermediate monocytes have different gene expressions in females versus males and one intermediate monocyte subset that may have predictive value in assessing the cardiovascular risk in females.
The text is well written and interesting. English is very good. I only have 2 minor suggestions:
The first paragraphs of Results contains information that should be moved to Methods.
I particularly find the gender differences you observed (iMo twice as high in females with high CVD compared to females with low CVD with no differences observed in males) to be interesting. Your findings may lead the way to more personalised care- maybe this could be listed as a strong point in the conclusion section. I feel the conclusions should be highlighted by including them in a separate section.
Round 2
Reviewer 1 Report
Comments and Suggestions for Authors
The terminology “cardiovascular disease” is not appropriated regarding the available information’s about the studied population. Again, since the population has been defined by coronary stenosis measured by Quantitative Coronary Angiography Cardiovascular disease should be replaced by coronary heart disease when speaking about the CAVA population.
Concerning the sex differences the authors did not answer my previous remark. Considering that specific subsets on genes are expressed in male vs female iMo and that there is a positive correlation between iMo and coronary stenosis severity in female but not males, it appears “artificial” to analyze the whole population after this being stated. I suggest then bringing information’s relative to sex differences at the end of the MS.
Round 3
Reviewer 1 Report
Comments and Suggestions for Authors
Following my remarks Chatterjee et al. updated the manuscript and moved all the sex-difference data to Figure 7. The MS seems more readable in this present form.
Comment 1.1
Section 4.1 “Since coronary heart disease and cardiovascular disease are closely related terms, we abbreviated them as CVD throughout this manuscript”.
You call patients population based on their phenotype / pathological traits, not because of “related terms”. Please be specific in the definition of your population to avoid confusion. CAVA patients have coronary heart disease/coronary stenosis/coronary artery disease as you prefer but not cardiovascular disease. In other words, you cannot extrapolate the current results to all cardiovascular diseases, but “just” to patients with a stenotic atherosclerotic lesion of the coronary arteries (which is already not so bad). These changes should include the text and the figures.
Comment 1.2
Figure 8 should come before Figure 7
Author Response
Please see attachement
